# Three-Dimensional Modeling with Osteoblast-like Cells under External Magnetic Field Conditions Using Magnetic Nano-Ferrite Particles for the Development of Cell-Derived Artificial Bone

**DOI:** 10.3390/nano14030251

**Published:** 2024-01-23

**Authors:** Chuang Ma, Makoto Izumiya, Hidehiko Nobuoka, Rintaro Ueno, Masaki Mimura, Katsuya Ueda, Haruka Ishida, Daihachiro Tomotsune, Kohei Johkura, Fengming Yue, Naoto Saito, Hisao Haniu

**Affiliations:** 1Institute for Biomedical Sciences, Interdisciplinary Cluster for Cutting Edge Research, Shinshu University, 3-1-1 Asahi, Matsumoto, Nagano 390-8621, Japan; 20hb403b@shinshu-u.ac.jp (C.M.); 21hb401k@shinshu-u.ac.jp (M.I.); 23bs217f@shinshu-u.ac.jp (H.N.); 23bs204d@shinshu-u.ac.jp (R.U.); 23bs223a@shinshu-u.ac.jp (M.M.); 19hb402j@shinshu-u.ac.jp (K.U.); haruka.i0729@gmail.com (H.I.); dtomo@shinshu-u.ac.jp (D.T.); yueratjp@shinshu-u.ac.jp (F.Y.); saitoko@shinshu-u.ac.jp (N.S.); 2Biomedical Engineering Division, Graduate School of Medicine, Science and Technology, Shinshu University, 3-1-1 Asahi, Matsumoto, Nagano 390-8621, Japan; 3Biomedical Engineering Division, Graduate School of Science and Technology, Shinshu University, 3-1-1 Asahi, Matsumoto, Nagano 390-8621, Japan; 4Department of Histology and Embryology, School of Medicine, Shinshu University, 3-1-1 Asahi, Matsumoto, Nagano 390-8621, Japan; kohei@shinshu-u.ac.jp

**Keywords:** nanoparticles, nano-ferrite, external magnetic field, osteoblast-like cells, artificial bone, bone regeneration, three-dimensional modeling, biocompatibility

## Abstract

The progress in artificial bone research is crucial for addressing fractures and bone defects in the aging population. However, challenges persist in terms of biocompatibility and structural complexity. Nanotechnology provides a promising avenue by which to overcome these challenges, with nano-ferrite particles (NFPs) exhibiting superparamagnetic properties. The ability to control cell positioning using a magnetic field opens up new possibilities for customizing artificial bones with specific shapes. This study explores the biological effects of NFPs on osteoblast-like cell lines (MC3T3-E1), including key analyses, such as cell viability, cellular uptake of NFPs, calcification processes, cell migration under external magnetic field conditions, and three-dimensional modeling. The results indicate that the impact of NFPs on cell proliferation is negligible. Fluorescence and transmission electron microscopy validated the cellular uptake of NFPs, demonstrating the potential for precise cell positioning through an external magnetic field. Under calcification-inducing conditions, the cells exhibited sustained calcification ability even in the presence of NFPs. The cell movement analysis observed the controlled movement of NFP-absorbing cells under an external magnetic field. Applying a magnetic field along the z-axis induced the three-dimensional shaping of cells incorporating NFPs, resulting in well-arranged z-axis directional patterns. In this study, NFPs demonstrated excellent biocompatibility and controllability under an external magnetic field, laying the foundation for innovative treatment strategies for customizing artificial bones.

## 1. Introduction

Artificial bone research and development advancements have garnered significant attention in medicine and bioengineering. The escalating human lifespan and the rising prevalence of fractures and bone defects [1] have underscored the necessity for highly efficient and biocompatible artificial bones [2,3,4]. Despite progress in conventional artificial bone materials, challenges, such as biocompatibility, structural complexity, and implant stability, persist and continue to demand innovative solutions [5].

Traditional approaches in bone tissue engineering commonly involve allografts or xenografts to reconstruct the structure and function of damaged bone tissue [2]. However, these approaches have inherent drawbacks. Allografts pose challenges due to their limited availability and potential for immune reactions, particularly concerning the quantity of harvested bone and the risk of unknown infections. To address these limitations, researchers have developed porous scaffolds composed of ceramic materials that mimic the inorganic component of bone [3,4]. Moreover, bioglass materials containing certain ions could also offer a suitable alternative [6]. While these materials, such as hydroxyapatite (HAp), offer promising structural support, they present significant challenges. Notably, HAp is non-biodegradable within the body and is not easily replaced by natural bone. Moreover, these ceramic materials lack the essential organic components of bone and live cells, leading to time-consuming repair processes [7,8].

In recent years, advancements in various 3D bioprinting technologies, including fused deposition modeling, stereolithography, binder jetting, and focused ion beams, have provided a fresh perspective on bone tissue engineering [9,10]. However, inherent flaws present potentially detrimental effects, such as droplet satellite formation, nozzle clogging, and issues related to extrusion pressure. Furthermore, ongoing research on the ideal bioink and reported cell survival rates falling below 90% pose challenges to 3D bioprinting’s full potential [11,12].

In this context, integrating nanotechnology in artificial bones is being considered. Nano-ferrite particles (NFPs), reduced to dimensions below several tens of nanometers, are known to exhibit superparamagnetism [13,14,15]. These particles can be measured for both bulk and single-particle magnetic properties using techniques such as the superconducting quantum interference device [13,16] and magnetic force microscopy [17], and they are recognized for demonstrating superparamagnetic behavior [13,14,15,18,19]. In a magnetic field, NFPs align their spins in the same direction and rapidly achieve saturation magnetization. Upon removal of the magnetic field, the spins become highly unstable and return to a demagnetized state due to thermal energy at room temperature, enabling the preparation of uniformly dispersed NFPs in a dispersing medium. Furthermore, NFPs typically demonstrate excellent biocompatibility, and their physical and chemical properties can be adjusted by varying their size, shape, and surface modifications. NFPs, as an attractive nanomaterial, are well-suited for various medical applications, including drug delivery systems (DDSs) [18], biosensing [19], and imaging [20], owing to their outstanding magnetic properties and biocompatibility [14,18,21,22]. These distinctive characteristics allow the induction of nanoparticles to specific target areas using a magnetic field [23]. Advancements in magnetic nanoparticles have facilitated the non-contact control of cell positioning using a magnetic field, offering novel possibilities for more effective therapeutic outcomes in bone tissue regeneration [18,21,22].

We have initiated the development of cell-derived artificial bones (CDABs) by utilizing osteoblasts (OBs) that uptake NFPs, constructing them in a 3D manner in any desired form and inducing calcification. This study explores the impact of NFPs on osteoblast-like cell lines (MC3T3-E1) and the efficient production and biocompatible cell-derived artificial bones using a magnetic field. By investigating NFP effects on MC3T3-E1 cells, this study seeks to unravel the potential role of NFPs. Furthermore, the anticipated induction of MC3T3-E1 cells incorporating NFPs using a magnetic field aims to control the shape of cell aggregates, fostering the development of biocompatible cell-derived artificial bones. This research is anticipated to provide new insights into the design and manufacturing of cell-derived artificial bones, with potential applications in fracture repair and healthcare implants.

## 2. Materials and Methods

### 2.1. Preparation and Sterilization of NFPs

NFPs were prepared using iron (II) chloride tetrahydrate (Hayashi pure chemical, Osaka, Japan) with a purity of 99% or higher, iron (III) chloride hexahydrate (Hayashi pure chemical) with a purity of 97% or higher, hydrochloric acid (Fujifilm Wako, Osaka, Japan) with a purity of 99% or higher, and sodium hydroxide (Fujifilm Wako) with a purity of 99% or higher. The traditional chemical method was employed to prepare the NFPs [14,15]. The procedure involved dissolving tetrahydrate iron chloride and hexahydrate iron chloride in 2 M hydrochloric acid. By adjusting the pH to be slightly acidic with sodium hydroxide, NFPs were precipitated. Non-nanoscale ferrite particles were removed using low-speed centrifugation at 1000× *g* for 20 min, followed by high-speed centrifugation to isolate the NFPs at 20,000× *g* for 30 min, which were then dissolved in ultrapure water. The prepared NFP solution was filtered through a 0.45 µm filter membrane for sterilization, followed by the additional separation of NFPs through high-speed centrifugation at 20,000× *g* for 30 min.

### 2.2. Preparation of Highly Dispersed NFPs

Fetal bovine serum (FBS; Gibco, Springfield, MO, USA) was diluted in ultrapure water and filtered through a 0.45 µm pore size filter membrane for sterilization to create a dispersant. The sterilized NFPs were then mixed with the dispersant to achieve a concentration of 50 mg/mL and were sonicated for 20 min using a PR-1 ultrasonic homogenizer (Thinky, Tokyo, Japan) and used as the baseline sample for each experiment. The rheological size and zeta potential of the dispersed NFPs were measured using a Zetasizer Nano ZS (Malvern Instruments, Malvern, UK), and the dispersed NFPs were adjusted to a concentration of 100 μg/mL and measured six times. Additionally, the NFPs were observed via a scanning electron microscope (SEM; JSM-7600F, JEOL Ltd., Tokyo, Japan) at 10.0 keV and transmission electron microscopy (TEM; JEM-1400HC, JEOL Ltd.) at 80.0 keV. Ninety particles were randomly selected, and their diameters were measured using FIJI ImageJ 2.14.0/1.54f (open-source project).

### 2.3. Cell Culture

The murine calvaria-derived osteoblast-like cell line (MC3T3-E1) was procured from Riken BRC (Tsukuba, Ibaraki, Japan). The MC3T3-E1 cells were cultured in alpha-minimum essential medium (αMEM; Nacalai Tesque, Kyoto, Japan) supplemented with 10% FBS and 1% penicillin-streptomycin (Fujifilm Wako) [24,25]. The passaging of cells was performed every three days, and the cells were cultured at 37 °C under 5% CO_2_.

### 2.4. Examination of Cell Proliferation

The MC3T3-E1 cells were seeded in 96-well plates at a density of 1.0 × 10^5^ cells/mL and cultured in αMEM containing 10% FBS and 1% penicillin-streptomycin. After 24 h of culture, NFPs, 100 µg/mL of ascorbic acid (Nacalai Tesque), and 5 mM of β-glycerophosphate (Calbiochem, LaJolla, CA, USA) were added to the culture for an additional 24 or 48 h. Cell proliferation was assayed using the alamarBlue^®^ reagent (Bio-Rad, Hercules, CA, USA). After 45 min of incubation with the diluted alamarBlue^®^ reagent (1:10 in DPBS), the fluorescence intensity was measured using PlateReader AF2200 (Eppendorf, Hamburg, Germany) at excitation/emission wavelengths of 535/590 nm [26]. The control group treated only with the dispersant was designated as 100%, and the cell proliferation was calculated relative to this control.

### 2.5. Evaluation of NFP Uptake

Under the same conditions described in Section 2.4, the MC3T3-E1 cells were seeded and cultured in µ-Slide 8 wells (ibidi GmbH, Martinsried, Germany). The cells were rinsed twice with DPBS after 24 h of NFP exposure to NFPs in MC3T3-E1 cells (NFP-MC) and stained with nuclear dye (H33342 fluorescent dye trihydrochloride in dimethyl sulfoxide solution; Nacalai Tesque, Kyoto, Japan) and lysosome staining solution (CytoPainter Lysosomal Staining ab138895; Abcam, Cambridge, UK) for 30 min [26]. Subsequently, the cells were washed twice with DPBS, and images were captured using a BZ-X710 inverted fluorescence microscope (Keyence, Osaka, Japan).

Furthermore, after incubation under the aforementioned conditions for 24 h, the cells were trypsinized, fixed using a 4% paraformaldehyde phosphate-buffered solution (4% PFA; Nacalai Tesque), centrifuged at 300× *g* for 10 min at room temperature, and the 4% paraformaldehyde phosphate-buffered solution was subsequently removed. The cells were resuspended in DPBS, and the side scatter light (SSC) was evaluated using a BD FACSCelestaTM flow cytometer (FCM; BD, Franklin Lakes, NJ, USA) [27,28,29].

### 2.6. Subcellular Localization of NFP-MC

Sterilized coverslips were placed in a 12-well plate, and the cells were cultured under the same conditions as described in Section 2.4. After 24 h of culture, the cells were exposed to 0.14 mg/mL NFPs. After 24 h of NFP exposure, the cells were washed with DPBS and fixed with 2.5% glutaraldehyde. The fixed cells were post-fixed, dehydrated, and embedded in epoxy resin using a 1% osmium tetroxide solution. Ultra-thin sections were prepared, and images were obtained using TEM.

### 2.7. Assessment of the Effects of Calcification on NFP-MC

MC3T3-E1 cells were seeded in a 24-well plate at a density of 4.0 × 10^4^ cells/mL and cultured in αMEM for 24 h. Subsequently, 100 µg/mL of ascorbic acid and 5 mM of β-glycerophosphate were added to αMEM to prepare a calcification medium. In the calcification medium containing the added NFPs, cells were exposed for 72 h after diluting the NFPs to concentrations ranging from 0.07 to 0.57 mg/mL; the group without NFPs served as the control group. Half of the medium was replaced with a fresh calcification medium without NFPs every 3 days. After 28 days of culture, the cells were fixed with 4% PFA for 20 min and stained with 1% Alizarin Red S (Sigma, St. Louis, MO, USA; pH 6.4) for 15 min to assess the calcification [24]. Following the staining process, the plates were washed and dried. To quantify, the calcified nodules were dissolved through agitation in a 5% formic acid solution for 10 min. The absorbance of the eluted dye was measured using PlateReader AF2200 at 405 nm.

### 2.8. Evaluation of the Uptake of NFPs under External Magnetic Field Conditions on MC3T3-E1 Cells Movement

MC3T3-E1 cells were seeded in a 24-well plate at a density of 4.0 × 10^4^ cells/mL and cultured in αMEM for 24 h. Ascorbic acid (100 µg/mL) and β-glycerophosphate (5 mM) were added to αMEM to prepare a calcification medium. In the calcification medium containing the added NFPs, cells were exposed to NFPs for 72 h after diluting the NFPs to concentrations ranging from 0.14 to 0.57 mg/mL. After washing the cells twice with DPBS, the cells were trypsinized. A 5 kOe external DC magnetic field was applied in-plane to the culture dish, and continuous images were captured using a BZ-X810 inverted fluorescence microscope (Keyence). The dynamic movement speed analysis was performed using BZ-X Analyzer Ver. 1.3.0.3 (Keyence).

### 2.9. Evaluation of the 3D Modeling of NFP-MC under Magnetic Field Conditions

MC3T3-E1 cells were seeded in a 10 cm culture dish at a density of 4.0 × 10^4^ cells/mL and cultured in αMEM for 24 h. The cells were exposed to NFPs at 0.14 mg/mL for 72 h. After washing the cells twice with DPBS, the cells were detached through trypsinization. A 5 kOe magnetic field was applied in the z direction to the µ-Slide III 3D Perfusion (ibidi GmbH). The cells were added to the µ-Slide, and αMEM calcification medium was used for 7 days with complete medium replacement every day. After 7 days of culture, the cells were gently washed with DPBS, followed by staining with nuclear dye. Simultaneously, the living cells were stained with CalceinAM (CellTraceTM Calcein Green AM, Thermo Fisher Scientific Jobs, Waltham, MA, USA) for 2 h. After staining, the cells were fixed with 4% PFA for 1 h, washed again with DPBS, sequentially immersed in 10% and 20% sucrose solutions (30% solution overnight), and embedded in OCT compound (Sakura Finetek Japan, Tokyo, Japan) to create 30 µm-thick consecutive cryosections using the Leica CM1950 cryostat (Leica Biosystems Inc., Buffalo Grove, IL, USA). Images were captured using a BZ-X710 inverted fluorescence microscope, and 3D image processing was performed using FIJI ImageJ 2.14.0/1.54f.

### 2.10. Statistical Analysis

The statistical analysis was conducted using GraphPad Prism Ver. 7.02 (Graph Pad Software Inc., San Diego, CA, USA). Significance tests were performed using Steel–Dwass or a one-way analysis of variance. Then, *t*-tests were used to compare the two groups and Tukey–Kramer or Dunnett tests were used for comparisons among multiple groups. The data were presented as mean ± standard deviation. A 95% confidence interval was assumed, and *p* < 0.05 was considered statistically significant.

## 3. Results

### 3.1. Properties of NFPs

The NFPs were meticulously assessed through SEM and TEM. The SEM image (Figure 1a) provides a surface-level view, while the TEM image (Figure 1b) offers a closer look at the particulate morphology of the NFPs. Quantitative data, summarized in Table 1, further elucidate the characteristics of the NFPs. As determined via the size distribution analysis (Figure 1c), the particle size was approximately 16.5 ± 2.4 nm, indicating a well-defined and uniform size distribution. Additionally, the hydrodynamic size, measured using Zetasizer Nano ZS, was approximately 164.7 ± 54.0 nm, signifying the effective dispersion of the NFPs in a liquid medium. The zeta potential, a key parameter reflecting particle stability, was noted as −9.9 ± 1.6 mV, indicative of a slightly negatively charged surface. These results collectively affirm the successful preparation of highly dispersed NFPs with precise morphological and physicochemical characteristics.

### 3.2. Effect of NFPs on the Cell Proliferation of MC3T3-E1 Cells

To investigate the impact of the NFPs on cell proliferation, MC3T3-E1 cells were exposed to different concentrations of NFPs. Figure 2 shows the concentration-dependent effect of NFPs on the cell proliferation of MC3T3-E1 cells. At lower concentrations of NFPs (0.07 and 0.14 mg/mL), no significant alteration in cell proliferation was observed during the 48 h and 72 h timeframes. At 0.28 mg/mL, there was a significant decrease in proliferation at 24 h (91%), with no significant difference detected at 48 h (91%), and a significant decrease in proliferation at 72 h (88%). However, when the MC3T3-E1 cells were exposed to a higher concentration of NFPs (0.57 mg/mL), a modest reduction in cell proliferation occurred, measuring at 90% after 24 h, 86% after 48 h, and 86% after 72 h. This decline in cell proliferation showed a concentration-dependent trend. At 0.57 mg/mL, the NFPs demonstrated a time-dependent effect, influencing cell proliferation, particularly at higher concentrations.

### 3.3. Intracellular Uptake of NFPs in MC3T3-E1 Cells

To investigate the cellular uptake of NFPs, MC3T3-E1 cells were exposed to a non-proliferation-affecting concentration (0.14 mg/mL), as illustrated in Figure 3. In Figure 3a–c, the control group without NFPs (Appendix A) demonstrated the localization of nuclear (H33342 fluorescent dye trihydrochloride in dimethyl sulfoxide solution) and lysosome (CytoPainter Lysosomal Staining ab138895) staining signals within the cells. Figure 3d provides the bright-field image of MC3T3-E1 cells exposed to NFPs for 72 h, showing NFPs (arrow) within the cellular area without clarifying their intracellular location (Appendix A). Figure 3f shows the merged image of Figure 3d,e, demonstrating the localization of nuclear and lysosome staining signals within the cells. Additionally, the position of the NFPs within the cell area shown in Figure 3d aligns well with the lysosome staining signal. However, as mentioned earlier, fluorescence microscopy observation and SSC alone do not provide conclusive evidence of whether NFPs are inside or outside the cells. To conclusively prove intracellular uptake, TEM was employed (Figure 3g,h). Importantly, in Figure 3h, NFPs were confirmed to be present only in the cytoplasmic region, not within the cell nucleus (indicated by the arrow).

### 3.4. Flow Cytometry Analysis of NFP Uptake in MC3T3-E1 Cells

The MC3T3-E1 cells were exposed to various concentrations of NFPs (0.07, 0.14, 0.28, and 0.57 mg/mL) for 72 h. The NFPs that internalized within the cells and adhered to the membrane were assessed by measuring the SSC values using FCM (Figure 4). The results revealed that as the concentration of NFPs increased up to 0.14 mg/mL, there was a corresponding increase in SSC values; this suggests that the elevated SSC in cells follows a concentration-dependent trend, which is associated with the increase in internalized and membrane-adhered NFPs. However, there was no significant change in the SSC values at NFP concentrations equal to or exceeding 0.28 mg/mL.

### 3.5. Effect of NFPs on MC3T3-E1 Cells under Calcified Conditions

To investigate the impact of NFP cellular uptake on cell functionality, MC3T3-E1 cells were exposed to various concentrations of NFPs (0.07, 0.14, 0.28, and 0.57 mg/mL) during calcification induction for 4, 8, and 12 weeks. After each period, MC3T3-E1 cells were stained with ARS (Figure 5a). Subsequently, the calcified nodules were dissolved in 5% formic acid for absorbance measurements (Figure 5b). Figure 5a shows the representative images of ARS staining for the MC3T3-E1 cells exposed to different concentrations of NFPs under calcification conditions. In the 4-week observation of the control group under calcification induction, calcified areas were observed. However, the groups exposed to NFPs at concentrations below 0.28 mg/mL showed a reduced calcified nodule area. Furthermore, the group exposed to 0.57 mg/mL NFPs showed a very minimal observable calcified nodule area. At 8 and 12 weeks, calcified nodule areas increased in all exposed groups. Figure 5b shows the absorbance measurement results of the dissolved calcified nodule solution after staining the calcified nodules of the MC3T3-E1 cells exposed to different concentrations of NFPs under calcification induction conditions. In the 4-week observation, all exposed NFP groups showed a significant decrease in absorbance compared to the control group. From the absorbance measurement results at 8 weeks, the values for each NFP-exposed group increased. The group exposed to 0.07 mg/mL NFPs after 8 weeks of calcification induction did not show a significant decrease compared to the control group. However, groups with NFP concentrations of 0.14 mg/mL and above showed a significant decrease. These absorbance measurement results demonstrated a concentration-dependent trend. From the quantitative results at 12 weeks, the values for each NFP-exposed group further increased. The measurement results showed no significant decrease compared to the control group in the groups exposed to NFP concentrations of 0.07–0.28 mg/mL.

### 3.6. Evaluation of Cell Movement in NFP-MC under a Magnetic Field

The cells incorporating NFPs at various concentrations were trypsinized, and their mobility in a planar direction under an external magnetic field of 5 kOe was evaluated in a non-adherent state. Randomly selected cells were used to measure their movement speed. Figure 6a shows representative images of the control group and MC3T3-E1 cells exposed to a concentration of 0.14 mg/mL NFPs when an external magnetic field was applied in the y direction. In the control group, minimal movement was observed at the start and 1 s after the magnetic field application. In contrast, MC3T3-E1 cells exposed to 0.14 mg/mL NFPs showed cell movement of approximately 2 µm in the y direction at the start and 1 s after the magnetic field application. Figure 6b presents the cell movement speed measurement results for the control and NFP groups. For each group, 20 cells were randomly selected. The results indicated that, due to the unadhered state of the cells in the control group, a movement speed of approximately 0.5 µm/s was observed. In the 0.07 mg/mL NFP concentration group, the cell movement speed increased to approximately 1.0 µm/s, showing a significant increase compared to the control group. In the 0.14 mg/mL NFP concentration group, the cell movement speed further increased to approximately 1.5 µm/s, demonstrating a significant increase compared to the control and 0.07 mg/mL NFP concentration groups. The cell movement speed in the 0.28 mg/mL NFP concentration group was approximately 1.4 µm/s, showing a significant increase compared to the control and 0.07 mg/mL NFP concentration groups. However, there was no significant increase compared to the 0.14 mg/mL NFP concentration group. In the 0.57 mg/mL NFP concentration group, the cell movement speed decreased to approximately 1.1 µm/s, showing a significant increase compared to the control group. However, it significantly decreased compared to the 0.14 mg/mL NFP concentration group. These results indicate that MC3T3-E1 cells exposed to NFP concentrations of 0.07 and 0.14 mg/mL showed an increasing trend in cell movement speed, with the maximum cell movement speed observed when exposed to concentrations of 0.14 and 0.28 mg/mL.

### 3.7. Evaluation of the 3D Modeling of NFP-MC under a Magnetic Field

To investigate the potential for further three-dimensional shaping, we applied a 5 kOe magnetic field along the direction perpendicular to the surface of the cultured cells for seven days. Afterward, we conducted staining and prepared continuous cryosections along the z-axis direction. Magnets were positioned above and below the culture chamber, applying a magnetic field perpendicular to the culture surface. Subsequently, NFP-MCs were introduced into the culture chamber, manipulated using the magnetic field, and arranged into a three-dimensional (3D) model based on the magnetic field orientation. Figure 7a shows a schematic diagram of the 3D modeling. Figure 7b depicts the cytoplasm of MC3T3-E1 cells stained with Calcein, while Figure 7c shows the cell nuclei of MC3T3-E1 cells stained with Hoechst. Even after seven days of culturing the constructs, both the cytoplasm and the nuclei of the cells were well-stained with a living cell marker, Calcein, indicating that the cells inside the constructs were normal. The results demonstrate that cells incorporating NFPs are arranged in a z-axis directional pattern under the influence of the magnetic field. Figure 7d presents the sequentially arranged consecutive cryosection images depicting the cells induced in a 3D pattern. The sequential fluorescence microscopy images of the cytoplasm demonstrate that NFP-MCs were induced and three-dimensionally modeled in response to the magnetic field (Appendix A).

## 4. Discussion

The current trend in artificial bones primarily involves ceramics [3,4,30]. However, natural bone matrices inherently contain organic components [31]. Recognizing this, we initiated this research with the perspective that using cultured cells is one viable approach to creating artificial bones with organic components. The experimental results collectively demonstrated the multifaceted interactions between NFPs and MC3T3-E1 cells, revealing promising implications for long-term cell culture and potential applications in artificial bone fabrication. At concentrations below 0.14 mg/mL, the NFPs exhibited minimal impact on MC3T3-E1 cells proliferation, suggesting favorable biocompatibility within this concentration range. Furthermore, the internalization of NFPs by MC3T3-E1 cells and their subsequent responsiveness to an external magnetic field highlighted the potential for controlled cellular movement. All the findings suggest the potential for precision shaping in preparing artificial bones with specific geometries.

Our study builds upon previous research on nano-magnetic materials based on iron oxide, reinforcing the established notion that such materials induce cellular uptake without compromising cell proliferation rates within a short period, as reported in various studies [32]. Even at the relatively high concentration of 0.57 mg/mL, our results demonstrate the remarkable maintenance of proliferation rates exceeding 80%, aligning with ISO 10993-5 standards [33] and ensuring biocompatibility.

The FCM results indicated an increase in cell complexity, suggesting the entry of iron oxide into the cells or its adhesion to the cell surface during measurements [32,34]. Fluorescence staining showcased a spatial correlation between NFPs and lysosomes, suggesting a potential role of lysosomes in NFP uptake [35,36]. Direct evidence from TEM confirmed the internalization of NFPs within the cytoplasm, even at concentrations where no observable impact on cell proliferation was noted. On the other hand, NFPs were distributed almost throughout the entire cell, as confirmed with the TEM image at a concentration of 0.14 mg/mL. Although no significant differences were observed in the SSC values at 0.14 mg/mL and above, noticeable differences in calcification ability and movement speed were identified; this suggests that SSC may not accurately reflect NFP uptake beyond a certain threshold (Appendix A). In fact, when evaluating the relative cellular uptake of nanomaterials using SSC, the concentrations of nanomaterials commonly used are often 0.1 mg/mL or lower [37,38]. Therefore, even if NFPs are further internalized into cells, the SSC value may not increase. In summary, the internalization of NFPs into cells appears to be concentration-dependent, and when exceeding a certain concentration, it may act as a contributing factor to the inhibition of cell proliferation, calcification, and movement speed.

Previous studies have reported that similar types of nano-magnetic materials suppress the differentiation of mesenchymal stem cells into OBs [14,39]. However, the literature on the impact of iron oxide-based nano-magnetic materials on bone formation is inconsistent. Some studies suggested enhanced differentiation of OBs, while others reported that, although the differentiation was promoted, bone formation was inhibited due to detachment from the substrate during a 20-day culture period [21]. The contradictory findings in differentiation inhibition are believed to depend on the particles’ composition, structure, and concentration [40,41]. In our study, the NFP-exposed groups maintained MC3T3-E1 cells functionality under calcification conditions for 8 and 12 weeks. We found no significant differences in calcification up to a concentration of 0.28 mg/mL compared to the control group; this suggests that the calcification induced with MC3T3-E1 cells tends to saturate over time, indicating the potential for artificial bone formation in long-term cultures. However, creating artificial bone using NFP-MCs may take longer than conventional 3D printing or hydrogel-based methods. Nevertheless, the method of creating 3D-forming cells without scaffold materials suggests the possibility of producing artificial bones containing cell-derived organic components.

Iron oxide nano-magnetic particles have been reported to label various cell types and exhibit the ability to move planarly under the influence of a magnetic field [36]. This experiment demonstrated that using a magnetic field allows for the positioning induction of NFP-OBs. Additionally, nano-magnetic particles, such as Ni and others, have been reported to undergo 3D induction under an external magnetic field, leading to self-assembly into nano-chains aligned with the vertical magnetic field [42]. Upon changing the direction of the external magnetic field, magnetic particles can be induced to take on desired shapes [43,44]. Our research observed the 3D formation of NFP-MC under the influence of a magnetic field, emphasizing the ability of the magnetic field to shape cells in the z-axis direction. The successful induction and maintenance of 3D cell arrangements represent a promising approach to 3D cell shaping through the strategic application of a magnetic field. This provides valuable insights for the potential development of custom cell-derived artificial skeletons. Furthermore, this precise control over spatial organization provides hope for the applications of tissue engineering and regenerative medicine.

## 5. Conclusions

In conclusion, our findings suggest the significant potential of NFPs in bone tissue engineering, impacting cell proliferation, internalization, calcification ability, and 3D modeling. The systematic exploration of the multifaceted interactions between NFPs and MC3T3-E1 cells not only unveils the potential applications of NFPs in bone tissue engineering and artificial bone synthesis but also provides a fresh perspective on using nanomaterials in the biomedical field. By comprehensively understanding the concentration-dependent effects, we can finely optimize the application conditions of NFPs, maximizing their potential in preparing artificial bones in the biomedical field. This innovation holds positive implications for promoting bone tissue regeneration and sparks innovative ideas for developing novel medical treatments and biomedical devices.

## Figures and Tables

**Figure 1 nanomaterials-14-00251-f001:**
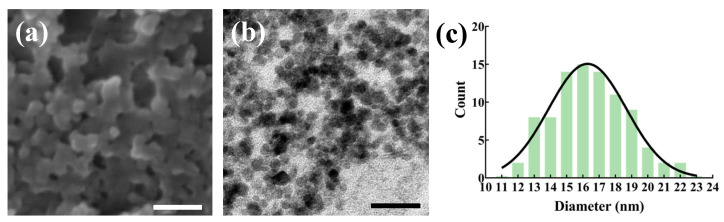
Results of the nano-ferrite observation. (**a**) SEM image of the observed nano-ferrite. (**b**) TEM image of the observed nano-ferrite. Scale bar: 50 nm. (**c**) NFP size distribution plot.

**Figure 2 nanomaterials-14-00251-f002:**
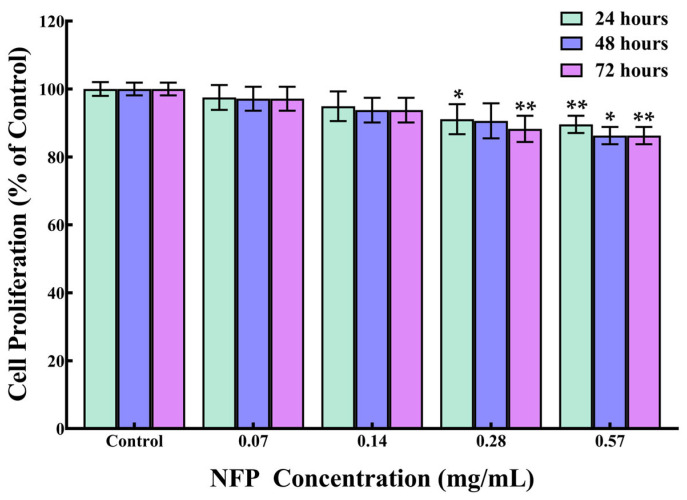
The impact of NFPs on the proliferation of MC3T3-E1 cells. Cell proliferation was analyzed at 24, 48, and 72 h after exposure to the NFPs. *n* = 4. The control group was exposed to the dispersant without the NFPs. The data are presented as mean ± standard deviation. The statistical significance was assessed using Dunnett’s multiple comparison test. *: *p* < 0.05, **: *p* < 0.01 compared to the control group.

**Figure 3 nanomaterials-14-00251-f003:**
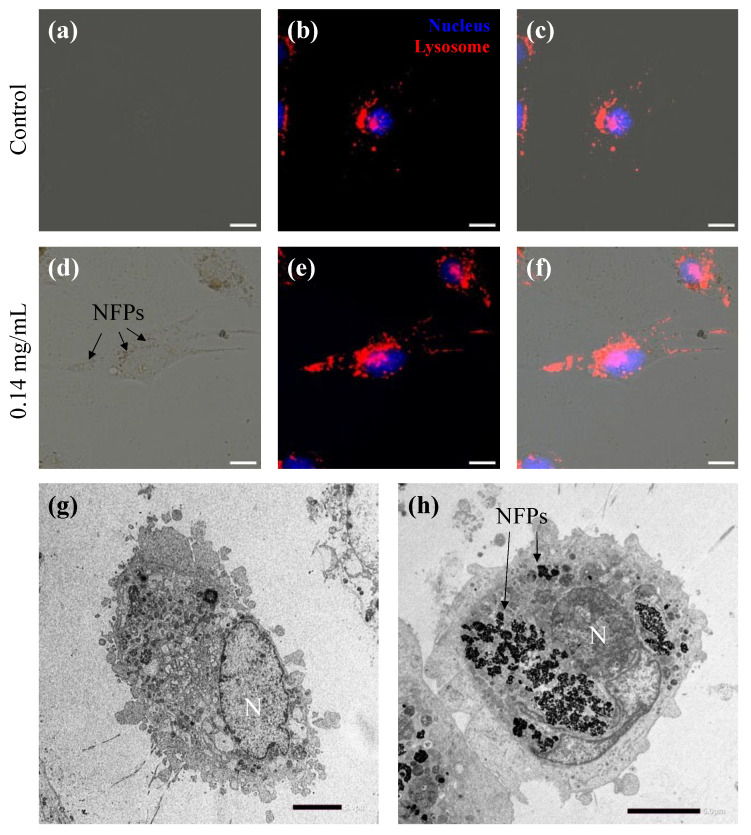
Cellular uptake of NFPs in MC3T3-E1 cells. (**a**) Bright-field image of the MC3T3-E1 cells control group culture without NFPs. (**b**) Fluorescently stained merged image of the MC3T3-E1 cells control group culture cells with H33342 for nuclear staining (blue) and the CytoPainter Lysosomal Staining Kit for the lysosomes (red). (**c**) Merged image of the images from (**a**,**b**). (**d**) Bright-field image of MC3T3-E1 cells exposed to 0.14 mg/mL NFPs for 72 h. The arrows indicate the NFPs. (**e**) Fluorescently stained merged image of MC3T3-E1 cells exposed to NFPs with nuclear and lysosome staining. (**f**) Merged image of images from (**d**,**e**). (**g**) TEM image of the MC3T3-E1 cells control group without NFPs. (**h**) The TEM image of MC3T3-E1 cells exposed to 0.14 mg/mL NFPs for 72 h indicates the intracellular localization of NFPs uptook by MC3T3-E1 cells. The arrows indicate NFP uptake within the cells. N shows the cell nucleus. The arrows indicate the intracellular NFP uptake within the cells. Scale bar: 5 µm.

**Figure 4 nanomaterials-14-00251-f004:**
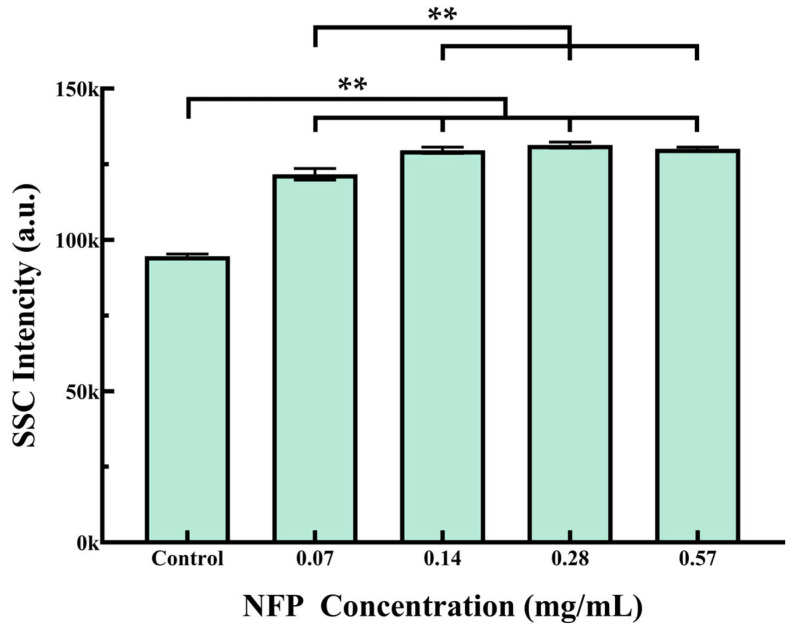
Flow cytometry analysis of NFP uptake in MC3T3-E1 cells. Cells were exposed to various concentrations of NFPs for 72 h. The control group was exposed to the dispersant without NFPs. The data are presented as mean ± standard deviation. The statistical significance was evaluated using Tukey’s multiple comparison test. **: *p* < 0.01.

**Figure 5 nanomaterials-14-00251-f005:**
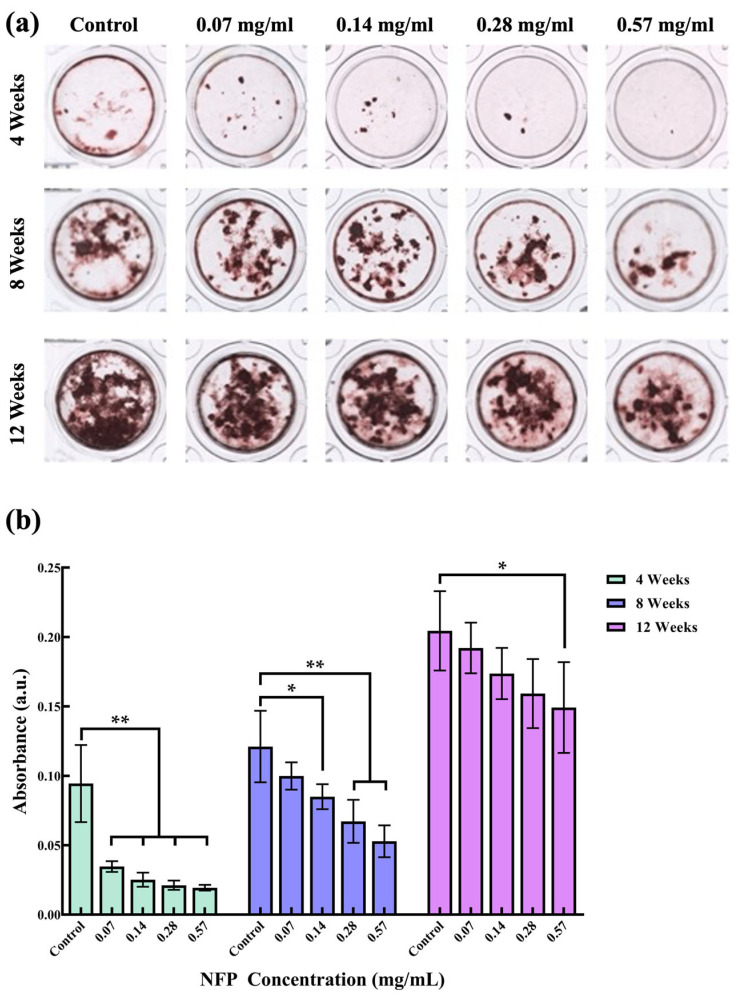
Calcification of MC3T3-E1 cells after 4, 8, and 12 weeks. (**a**) Representative images of Alizarin Red S (ARS) staining in various culture media under calcification conditions at 4, 8, and 12 weeks. (**b**) Absorbance measurements and quantification at 405 nm of the mineralized nodule solution washed with 5% formic acid for 10 min. *n* = 4. The data are presented as mean ± standard deviation. The statistical significance was evaluated using Dunnett’s multiple comparison test. *: *p* < 0.05, **: *p* < 0.01 compared to the control group.

**Figure 6 nanomaterials-14-00251-f006:**
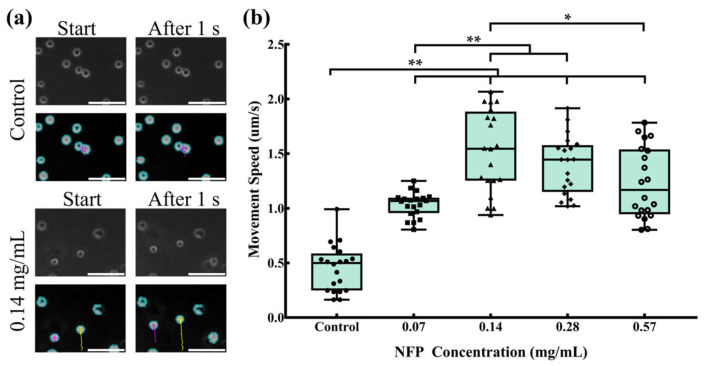
Movement speed of MC3T3-E1 cells. Impact of NFPs on the movement speed of MC3T3-E1 cells. (**a**) Typical images of cell movement for the control and 0.14 mg/mL groups. The upper figure shows phase-contrast images, while the lower figure illustrates cell-tracking images analyzed with the BZ-X analyzer. Scale bar: 100 µm. (**b**) Analysis of the movement speed of cells exposed to various concentrations of NFPs. The control group was exposed to the dispersant without NFPs. The data are presented as mean ± standard deviation. The statistical significance was assessed using Tukey’s multiple comparison test. *: *p* < 0.05, **: *p* < 0.01.

**Figure 7 nanomaterials-14-00251-f007:**
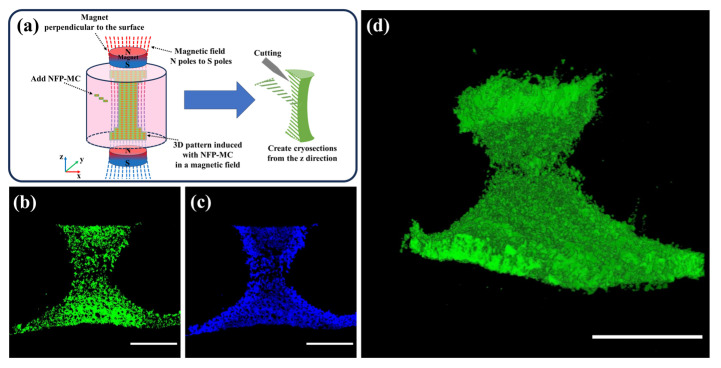
Evaluation of NFP-MCs for three-dimensional modeling under a magnetic field. (**a**) Schematic diagram during the three-dimensional modeling and cryosection preparation. (**b**,**c**) Cytoplasmic (green) and nuclear (blue) staining of NFP-MCs. (**d**) Observation of the three-dimensional cell models after the sequential arrangement of the cryosection images. Scale bar: 1000 µm.

**Table 1 nanomaterials-14-00251-t001:** NFP size and zeta potential.

Materials	TEM-Observed Particle Size (nm)	Rheological Size (nm)	Zeta Potential (mV)
NFPs	16.5 ± 2.4	164.7 ± 54.0	−9.9 ± 1.6

## Data Availability

All data are available from the corresponding author upon reasonable request.

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
