# Peer review of "Three-Dimensional Modeling with Osteoblast-like Cells under External Magnetic Field Conditions Using Magnetic Nano-Ferrite Particles for the Development of Cell-Derived Artificial Bone"

_nanomaterials, 2024, doi:10.3390/nano14030251_

Round 1
Reviewer 1 Report
Comments and Suggestions for Authors
This study aims to investigate the biological effects of nano-ferrite particles (NFPs) on osteoblast-like cell. Several experiments, including cell viability, NFPs uptake of, calcification and cell migration were tested. Overall, since these results can be found in several previous reports, the novelty of these parts are low. However, there is also some parts show interesting results for the reader. The experiment shows that when applying a magnetic field along the z-axis can induces three-dimensional shaping of the cells in z-axis directional patterns. This result seems the first-time to present in literature. The reviewer has the following comments for the authors to modify their report:
1. This article used twenty parties to calculate the average diameter of the produced NPs. It is not reasonable because the twenty parties in image can be selected. The author should provide a figure to show up the real particle size distribution.
2. Fig. 2, only 24 and 48 hrs results are provided in the figure. It is not reasonable because all the following experiments were performed for 72 hrs or much longer. Since the authors concluded that “At concentrations below 0.14 mg/mL, NFPs exhibited minimal impact on MC3T3-E1 proliferation, suggesting a favorable biocompatibility…”, the author should provide a complete cell growth curve graph for this concentration.
3. The part regarding cells arrange along z-axis and become three-dimensional shape is interesting. However, the discussion of this result is only one sentence. I strongly suggest the author to provide a new paragraph to discuss this part. Without this, this article become meaningless because other part can be found in previous references.
Reviewer 2 Report
Comments and Suggestions for Authors
The manuscript titled “Three-Dimensional Modeling with Osteoblast-Like Cells under External Magnetic Field Conditions Using Magnetic Nano-Ferrite Particles for the Development of Cell-Derived Artificial Bone” by Ma, C.; et al. is a scientific work where the authors study the positive effect of the addition of ferrite magnetic nanoparticles seeding in MC3T3-E1 osteoblast cells. The results shown a good biocompatibility with low cellular citotoxicity enabling the cell proliferation. Incubation times of 24 h and 48 h were tested in this regard. Then, fluorescent microscopy images revealed the precisely control of the nanoparticle movement under the exposition of external magnetic fields. The data is interesting and the manuscript is generally well-written.
However, it exists some points that need to be addressed (please, see them below detailed point-by-point). The most relevant outcomes remarked by the authors can contribute in the growth of many fields like the design and development of the next-generation of scaffolds for bone tissue engineering. For this reason, I will recommend the present scientific manuscript for further publication in Nanomaterials all the below described suggestions will be properly fixed.
Here, there exists some points that must be covered in order to improve the scientific quality of the manuscript paper:
1) KEYWORDS (OPTIONAL). The authors should consider to add the term “external magnetic field” (or “magnetic nanopositioning”) in the keyword list.
2) INTRODUCTION. This section perfectly outlines the state-of-the art in this field. Please, the authors can find the following points to strengthen this content. Firstly, it may be advisable to provide some quantitative information about the worldwide burdens related to human bone fractures [1] in order to make the potential readers aware about the importance of this topic.
[1] GBD 2019 Fracture Collaborators. Global, regional, and national burden of bone fractures in 204 countries and territories, 1990-2019: a systematic analysis from the Global Burden of Disease Study 2019. Lancet Healthy Longev. 2021, 2, e580-e592. https://doi.org/10.1016/s2666-7568(21)00172-0.
3) “Traditional approaches in bone tissue engineering commonly involve allografts or xenografts to reconstruct the structure of damaged bone tissue. However (…) To address these limiations, researchers have developed porous scaffolds made from ceramic materials that mimic the inorganic component of bone (…) time-consuming repair processes (page 2). I agree with this statement provided by the authors. In this framework, the use of glass materials containing certain ions could also offer a suitable alternative [2] (In addition to the aforementioned ceramic scaffolds).
[2] Gavinho, S. R.; et al. Bioactive Glasses Containing Strontium or Magnesium Ions to Enhance the Biological Response in Bone Regeneration. Nanomaterials 2023, 13, 2717. https://doi.org/10.3390/nano13192717.
4) “In recent years, advancements in 3D bioprinting (…) challenges to 3D bioprinting’s full potential” (page 2). Here, the authors should list all the existing 3D bioprinting technologies (e.g. fused deposition modeling, stereolithography, binder jetting, among others based on direct energy deposition like focused ion beam). Then, in the limitation part the authors should not neglect the potential detrimental effects linked to the droplet satellite formation and nozzle clogging.
5) “Nano-ferrite particles (NFPs), reduced to dimensions below several tens of nanometers are known to exhibit superparamagnetism (…) using magnetic field, offering novel possibilities for more effective therapeutic outcomes in bone tissue regeneration” (page 2). Finally, the authors should also introduce bulk [3] and single molecule [4] techniques to characterize the intrinsic nanoparticle magnetic properties.
[3] Calvo, R.; et al. Novel Characterization Techniques for Multifunctional Plasmonic-Magnetic Nanoparticles in Biomedical Applications. Nanomaterials 2023, 13, 2929. https://doi.org/10.3390/nano13222929.
[4] Winkler, R.; et al. A Review of the Current State of Magnetic Force Microscopy to Unravel the Magnetic Properties of Nanomaterials Applied in Biological Systems and Future Direction for Quantum Technologies. Nanomaterials 2023, 13, 2585. https://doi.org/10.3390/nano13182585.
6) MATERIALS & METHODS. “Additionally, NFPs were observed via a scanning electron microscope (…) and transmission electron microscopy (…) FIJI ImageJ 2.14.0/1.54f” (page 3). The acceleration voltage (in keV) employed in this research should be detailed in this subsection.
7) “2.6. Subcellular Localization of NFP-MC (…) Ultra-thin sections were prepared, and images were obtained using TEM” (page 4). Did the authors observe any damage cause by the electron exposition during the TEM data acquisition (even if the examined cells are stiff)? A brief statement should be provided in this regard.
8) “2.9. Evaluation of 3D Modeling of NFP-MC Under Magnetic Field Conditions (…) Image were captured using a BZ-X710 inverted fluorescence microscope, and 3D image processing was performed using FIJI ImageJ 2.14.0/1.54.f” (pages 4-5). Did the authors experienced any photobleaching effect during the data acquisition which could impact on their subsequent interpretation? Some discussion should be furnished in this regard. In case affirmative, what where the strategy actions followed by the authors in order to minimize this negative effect?
9) RESULTS. “Additionally, the hydrodynamic size, measured using Zetasizer NanoZS, is approximately 164.7 ± 54.0 nm, signifying the effective dispersion of NFPs in a liquid medium” (page 5). Thus, no aggregation effects were observed when the nanoparticles are suspended in liquid environments?
10) Figure 3, panels a) and d) (page 7). Did the authors take some phase contrast illumination images from the same areas? The depicted bright-field images are dim to unequivocally reveal the observed feature surroundings. This information could be placed as Supplementary Information (SI). Same comment for the Fig. SI 2.
11) DISCUSSION. This section perfectly remarks the most relevant outcomes found the the authors providing a fruitful discussion for all the potential readers. No actions are requested from the authors.
12) CONCLUSIONS. Here, it may be advisable if the authors could highlight some future lines to pursue this research. Finally, the references are in the proper format of Nanomaterials journal (No actions are requested from the authors in this regard).
Comments on the Quality of English LanguageThe manuscript is generally well-writtein albeit it may be advisable if the authors take a final check in order to polish those final details susceptible to be improved.
Round 2
Reviewer 1 Report
Comments and Suggestions for Authors
The authors have revised their manuscript according to my comments. It can be accepted in current form.